# Differences and Similarities in Predictors of Expressive Vocabulary Development between Children with Down Syndrome and Young Typically Developing Children

**DOI:** 10.3390/brainsci11030312

**Published:** 2021-03-02

**Authors:** Kari-Anne B. Næss, Johanne Ostad, Egil Nygaard

**Affiliations:** 1Department of Special Needs Education, University of Oslo, 0318 Oslo, Norway; 2The National Library of Norway, 0255 Oslo, Norway; johanne.ostad@nb.no; 3Department of Psychology, University of Oslo, 0317 Oslo, Norway; egilny@psykologi.uio.no

**Keywords:** Down syndrome, expressive vocabulary, language development, predictors, longitudinal study

## Abstract

The purpose of this study was to examine potential differences in the predictors of expressive vocabulary development between children with Down syndrome and typically developing children to support preparation for intervention development. An age cohort of 43 children with Down syndrome and 57 typically developing children with similar nonverbal mental age levels were assessed at three time points. Linear mixed models were used to investigate the predictors of expressive vocabulary over time. Both groups achieved progress in expressive vocabulary. The typically developing children had steeper growth than the children with Down syndrome (1.38 SD vs. 0.8 SD, *p* < 0.001). In both groups, receptive vocabulary, auditory memory, and the home literacy environment were significant predictors of development. In the children with Down syndrome, the phonological awareness and oral motor skills were also significant. Group comparisons showed that receptive vocabulary, auditory memory and oral motor skills were stronger predictors in the children with Down syndrome than in the typically developing children. These results indicate that children with Down syndrome are more vulnerable when it comes to risk factors that are known to influence expressive vocabulary than typically developing children. Children with Down syndrome therefore require early broad-based expressive vocabulary interventions.

## 1. Introduction

Vocabulary can be defined as an individual’s stock or body of words [1]. A distinction is usually made between the modalities of receptive and expressive vocabulary, which refer to the comprehension and production of words, respectively. Weak expressive vocabulary skills are a well-known correlate of reduced academic, behavioral, and social outcomes [2,3,4,5,6,7], whereas well-developed skills support the expression of one’s feelings, meanings and knowledge and positively influence autonomy [8] as well as emotional and behavioral functioning [9].

One group showing especially weak expressive vocabulary skills and in critical need of intervention is children with Down syndrome [10]. However, there are uncertainties regarding their developmental profile as well as the predictors of development, and few intervention programs aimed at expressive vocabulary in this group of children are available that have been tested through research [11,12].

The present study therefore seeks to identify potential predictors of expressive vocabulary development in a national age cohort of children with Down syndrome compared to those of young typically developing children. The objective in doing so is to reveal knowledge essential for planning appropriate expressive vocabulary interventions especially adapted for the language profile of children with Down syndrome.

## 2. Expressive Vocabulary, a Specific Area of Weakness in Children with Down Syndrome?

Down syndrome, usually caused by an extra chromosome 21, is the single largest cause of intellectual disability and the most common birth disorder [13]. It is found to cause an overall delay in development, albeit with great variation between individuals [14].

Language is one of the most impaired domains of functioning among children with Down syndrome [15]. Their expressive vocabulary has often been shown as a weakness relative to their own receptive vocabulary and to the expressive vocabulary of typically developing children of the same nonverbal mental age level as assessed by clinical tests [16,17,18,19]. However, some studies focusing on very young children and applying parental reports have found no significant differences in expressive vocabulary between children with Down syndrome and typically developing children (e.g., [20,21,22,23]). In contrast, other studies that used naturalistic language samples had similar findings as studies that used standardized tests, showing that children with Down syndrome had a more limited vocabulary breadth; they used fewer words and showed less variety in the words they used than typically developing children matched for nonverbal mental age [24,25].

Although a large variation in vocabulary development among children with Down syndrome is recognized and although their profile may differ from that of typically developing children according to children’s age or the measures used, some general milestones have been suggested in their expressive vocabulary development (see Table 1).

**Table 1 brainsci-11-00312-t001:** Overview of the main stages of expressive vocabulary development in young typically developing children and children with Down syndrome.

Stages of Early Expressive Vocabulary Development	TD	DS
Developmental process	Rapid growth during early childhood [25,26,27]	Slow growth during childhood [17,28]
Utterance of first words	At the age of approximately 1 year [29,30]	At the age of approximately 1–9 years [31], but large individual variation [32]. Some say their first words at the same time as TD children [33]
Achievement of 50 words	At the age of approximately 18 months [32,34]	For 25%, at approximately 3 years of age; for 50%, at approximately 4 years; and for 75%, at approximately 5 years [32]. Large individual variation [35]
Onset of sentence production, i.e., combination of two words or more	At the age of approximately 18 months [36] or after reaching 50 words [34,36]	Approximately 3.5–5 years of age [37,38] or after reaching 100 words [39]

Note. TD = typically developing children; DS = children with Down syndrome; regarding the onset of sentence production, the ties between maturation, linguistic, and/or cognitive development have been discussed [40].

The pace of vocabulary growth varies [41], and there are within group-differences both in children with Down syndrome and in typically developing children, but in general, it becomes apparent from Table 1 that the vocabulary development in children with Down syndrome differs from the development in typically developing children at all main stages. Notably, there is some inconsistency across studies as to whether the development of expressive vocabulary and language learning in children with Down syndrome are delayed as a result of a general intellectual disability (c.f. [42]) or are deviant and thereby related to a specific Down syndrome phenotype (c.f. [43]). However, both delayed and deviant development imply a difference from typical development and thereby may raise questions about whether the same predictors of expressive vocabulary exist in children with Down syndrome and typically developing children.

### 2.1. Predictors of Expressive Vocabulary Development

Both environmental and child-related (individual) variables are found to reliably predict the development of expressive vocabulary in typically developing children (c.f. [27,44,45]). Environmental variables that earlier research has shown to be related to typical expressive vocabulary development are socioeconomic status, usually based on measures of the family’s income, education and/or occupation [46,47,48], if the child is read regularly with/to [49,50,51], and family history of speech/language problems [52,53,54].

Child-related variables that have been found to be related to expressive vocabulary in typically developing children are nonverbal mental age [55,56], gender [53,54,57,58], socioemotional functioning [59], receptive vocabulary [60], verbal short-term memory [61], phonological awareness [62,63], oral motor skills [64], and reading [65,66].

It is often assumed that the same predictors apply to children with Down syndrome (e.g., [67]), but limited research exists, and clinical test results of vocabulary measures of an age cohort of children with Down syndrome from different time points have not been compared to those of typically developing children. However, Deckers et al. [67] investigated the predictive role of individual and environmental variables for expressive vocabulary breadth development using a parental survey and a standardized clinical test in a sample of 30 children with Down syndrome aged between 2–7 years and compared their results to norms for typically developing children. Based on the delayed hypothesis, they presumed that the same predictors of expressive vocabulary development for typically developing children would also apply to children with Down syndrome. This assumption was confirmed in their study.

In total, very few longitudinal studies of the expressive vocabulary development of children with Down syndrome have been conducted (c.f. [68]), and the existing studies were small-scale studies or did not include clinical test data, which limited the multifactorial statistical analysis of predictor variables. Thus, little is known about the predictors of expressive vocabulary in children with Down syndrome and whether these predictors are similar or different from those in typically developing children. This knowledge is of importance to support preparation for expressive vocabulary intervention development specifically adapted to children with Down syndrome.

### 2.2. The Current Study

To summarize, there is a critical need to determine whether the predictors of expressive vocabulary in young children with Down syndrome are the same as those in typically developing kindergarten children to provide the knowledge needed to develop effective vocabulary interventions for children with Down syndrome. The research questions to guide this exploratory study are as follows:What variables at 3 years of age predict expressive vocabulary in typically developing children at ages 3, 4 and 5 years?What variables at 6 years of age predict expressive vocabulary in children with Down syndrome at ages 6, 7 and 8 years (first three years of primary school)?What differences and similarities exist between the estimated effects of the predictors of expressive vocabulary between children with Down syndrome and typically developing children?

## 3. Materials and Methods

### 3.1. Participants

The present study was approved by the Norwegian Regional Committees for Medical and Health Research Ethics, and the invitation letter and consent form were sent out through the Norwegian National Habilitation Service to the parents of an age cohort of Norwegian-speaking children with Down syndrome. The parent(s) who wanted their child to take part in the study had to fill out, sign and send in the consent form. This recruitment procedure resulted in 43 children with Down syndrome (mean chronological age = 75.78 months, SD = 3.48 months, range = 69.18–81.11 months, mean nonverbal mental ability raw score (Block Design test from the Norwegian version of the Wechsler Preschool and Primary Scale of Intelligence III (WPPSI-III) [69]) = 12.23, *SD* = 5.40) and one of their parents participating. Based on the inclusion criteria for the participants with Down syndrome stated in the invitation letter, the children had to be six years of age, have no comorbid diagnoses of autism and have at least one parent who spoke Norwegian as his/her first language. A control group of 57 typically developing children initially matched for nonverbal mental ability (mean chronological age = 36.67 months, *SD* = 4.17 months, range = 29.11–44.18 months; mean nonverbal mental ability raw score = 12.61 (Block Design), *SD* = 4.63) and one of their parents were recruited from eight different kindergartens in a typical Norwegian municipality. To be included in the control group, the children had to have Norwegian as his/her first language and no history of special educational needs. There were initially no significant differences between the two groups of children based on nonverbal mental ability as measured by raw scores from the Block Design test (t(98) = 0.38, *p* = 0.71, *d* = 0.08).

The gender distribution was quite similar across groups (OR (100, 1) = 1.09, *p* = 1.00), with 21 girls (48.8%) in the group with Down syndrome and 29 girls (50.9%) in the typically developing group. See Table 2 for more descriptive information about the samples.

### 3.2. Data Collection Procedure

A dual approach to data collection, including clinical tests of the children and parental questionnaires, was used.

The clinical tests were administered individually in kindergartens or schools with approximately 12 months between test points (T1, T2 and T3). The test answers were registered manually, and expressive answers were additionally audiotaped.

The parental questionnaires were sent digitally via email to one parent of each participating child. If answers were not received by the deadline, up to two reminders were sent.

### 3.3. Measures

We used standardized tests that are commonly used in research and that have satisfactory validity and reliability. In general, standardized procedures, including specified starting points and discontinuation rules, were followed for the administration of all tests. In the scoring process, the expressive test scores were summed without penalty for systematic articulation errors.

### 3.4. Expressive Vocabulary

The dependent variable, expressive vocabulary, was assessed with the Picture Naming test (from the Norwegian version of the WPPSI–III; [69]). For this 38-item naming test, the child was shown one picture at a time and had to name it verbally. The total test score corresponded to the number of correct answers.

### 3.5. Receptive Vocabulary

Receptive vocabulary was assessed using the British Picture Vocabulary Scale 2nd edition (BPVS-II; [70]; adapted to Norwegian by Lyster, Horn & Rygvold [71]). For this 144-item comprehension test, the child was shown four pictures and had to point to the picture that corresponded to the stimulus word said by the examiner. The total test score corresponded to the number of correct answers.

### 3.6. Auditory Memory

The mean of the standardized scores from tests of the three following phonological processing skills were used as a measure of auditory memory: verbal short-term memory, sentence memory, and nonword repetition.

Verbal short-term memory was assessed using Word Span [72], which is a 24-item auditory test. In this test, children heard a list of spoken words that they had to repeat in the correct order. The total test score corresponded to the number of correct answers (all words had to be said in the correct order).

Sentence memory was assessed by the Sentence Repetition test (from the Norwegian version of WPPSI–R; [73]), which is a 21 auditory repetition test. In this test, children heard sentences that they had to repeat correctly. The total test score corresponded to the number of correct answers.

**Nonword repetition** (original by [45]; adapted to Norwegian by Furnes & Samuelson [74]) was assessed with a 28-item word repetition test. For each item, children heard a nonword, which they had to repeat. The nonwords varied in length from two to five syllables. The total test score corresponded to the number of correct answers.

### 3.7. Phonological Awareness

The mean standardized scores from four tasks by Carroll et al. ([62]; adapted to Norwegian by Child Language and Learning [72]) were used as a measure of phonological awareness. The rhyme and phoneme tasks are each 16-item implicit measures, and the initial and final syllable tasks are each 8-item implicit measures. All four tasks use a similar framework. For each item, two pictures were laid on a table in front of the child, and a third picture was shown to the child by a puppet. For example, for the phoneme measure, a Benny Beaver the puppet had a picture of a house, and the tester asked the child, “Which of these words, sock or hat, starts with the same phoneme as house?” The task did not require an oral reply.

### 3.8. Oral Motor Skills

Oral motor skills were assessed using the Nordic Orofacial Test-Screening (NOT-S) [75]. The NOT-S is a 17-item screening instrument for 6 orofacial function domains that includes the evaluation of the face at rest, nose breathing, mimicry, chewing muscle and jaw function, oral motor skills, and speech. One or more “yes” responses to items within a domain indicate orofacial dysfunction. The present study excluded three items related to pronunciation; thus, oral motor skills were measured as the mean score of 14 items (0 = no, 1 = yes).

### 3.9. Nonverbal Mental Ability

Nonverbal mental ability was assessed using the Block Design subtest from the Norwegian version of the WPPSI-III [69]. For this 20-item visual-spatial and organizational processing problem-solving test, the child was shown a model with building blocks (the first 13 items) or a picture (items 13–20), which they had to copy. To calculate the total test score, 2 points were given per correct answer (items 1–6 allowed two attempts, and 1 point was given if the answer was correct on the second attempt).

### 3.10. Parental Questionnaires

The following digital surveys were sent via email to one parent per child. Up to two reminders were sent if the answer was not received.

### 3.11. Socioemotional Functioning

Socioemotional functioning was assessed using the Strengths and Difficulties Questionnaire (SDQ) [76]. The SDQ is a 25-item parental statement questionnaire used to measure children’s social problems and social capabilities. It is separated into four subfactors measuring problems (emotional symptoms, conduct problems, hyperactivity/inattention, and peer relation problems) and one subfactor measuring capabilities (prosocial behavior). The total score was calculated as the sum of the scores of 20 items from the four problem-oriented subfactors that could be answered on a three-point scale (0–2), from not true to definitely true (possible range 0–40). The scoring of five statements phrased in opposite direction of related items was reversed before the calculation of the total score.

### 3.12. Background Questionnaire, Socioeconomic Status (SES)

SES was assessed using an electronic parental background questionnaire. SES was assessed based on the parents’ highest education level achieved, with 0 indicating elementary school only and 4 indicating 4 years or more of university-level education. A summary mean score of the mother’s and father’s level of education was used.

### 3.13. Background Questionnaire, Home Literacy Environment

The home literacy environment was assessed in the parental background questionnaire as described above and was based on one question regarding the frequency of the parents reading aloud to the child, with the alternatives once a month or less (1), several times a month (2), approximately once a week (3), several times a week (4) and every day or almost every day (5).

### 3.14. Background Questionnaire, Reading Skills

Reading skills were assessed in the parental background questionnaire based on the following question: “How many words can the child read?” The answer options were 1–5 (1), 6–10 (2), 11–15 (3), 16–20 (4) and above 20 (5). 

### 3.15. Background Questionnaire, Language Problems in the Family

Language development in the family was assessed by one question on the parental background questionnaire. Language problems in the family was coded as 1, and no language problems in the family was coded as 2.

### 3.16. Data Analyses

We used IBM SPSS statistics version 26 for all analyses. The significance level was set at 0.05, and all tests were two-tailed.

Linear mixed models were used to investigate the predictors of expressive vocabulary over time. To take into consideration the risk of both type I and type II errors we used (a) Bivariate models; (b) Models including a limited number of confounders selected a priori. The choice of covariates was in line with VanderWeele [77] guided by empirical evidence/theoretical knowledge of established confounding variables for both expressive language skills and the included predictors. The covariates were the sociodemographic background variable parental education (e.g., [78,79]), gender (e.g., [80,81]), and the g-factor of nonverbal mental abilities that is hypothesized to impact learning (e.g., [82]); (c) A full model with all predictors. entered as fixed effects: gender, nonverbal mental abilities, parental education, home literacy environment, language problems in the family, socioemotional functioning, auditory memory, oral motor skills, phonological awareness, receptive vocabulary, and number of words the child can recognize. The dependent variable was expressive language over time, as measured with the Picture Naming test at three different times. Time was therefore included in the model as a fixed effect, with the Picture Naming test being the only repeated measure. All predictors were measured at the first time point. All models included the two-way interaction terms between the group and predictor variables, thus analyzing whether the difference in the regression coefficient between the groups was significant. All models were rerun with the groups reversed to determine the regression coefficient of both groups. All continuous variables were standardized (z-values) before being entered into the model; thus, the presented estimates (regression coefficients (*b*)) resemble Cohen’s *d* values. The two original problem-oriented measures (socioemotional functioning and oral motor skills) were reversed before being entered into the mixed models.

We performed preliminary analyses of the relevant assumptions for linear mixed models through visual interpretation of predicted probability (P-P) plots and scatterplots of predicted values and residuals. The assumptions of homoscedasticity and normally distributed residuals, and thus linearity, were fulfilled for all models. There was no indication of high collinearity, with no correlations between predictors above 0.55 and VIFs between 1 and 2.5. Little’s MCAR test indicated that data were not missing completely random (χ^2^ = 91.0, *p* = 0.02), with a mean of 1.7% of the values being missing. We used the restricted maximum likelihood (REML) estimation method, with missing replaced by Expectation Maximation (EM) imputation. Thus, we did not exclude any existing information in the mixed models [83].

## 4. Results

### 4.1. Descriptive Statistics

The gender distribution was quite similar across groups (OR (100, 1) = 1.09, *p* = 1.00), with 21 girls (48.8%) in the group with Down syndrome and 29 girls (50.9%) in the typically developing group. There were no major differences between groups in how often the parents read to their children (22% vs. 19% once a week or more seldom, 33% vs. 19% several times a week, and 45% vs. 63% every day or almost every day for the children with Down syndrome vs. the typically developing children, respectively, chi-square (96, 4) = 7.94, *p* = 0.09). The proportion of families with known language problems or dyslexia was also quite similar across groups (OR (98, 1) = 1.16, *p* = 0.81), with 21% and 24% of families having such problems in the groups of children with Down syndrome and typically developing children, respectively. However, the children with Down syndrome recognized more words than the typically developing children (chi-square (98, 4) = 23.54, *p* < 0.001). While 67% of the typically developing children could not recognize any words, most of the children with Down syndrome recognized 1–5 words (47%), and some of the children with Down syndrome recognized 6–10 words (16%), 11–15 words (5%) or more than 20 words (7%). Twelve (28%, missing information for two) of the children with Down syndrome had had chronic hearing loss, whereas none of the typically developing children were reported to have had such problems (missing information for two children). A majority (*n* = 29, 67%) of the children with Down syndrome used alternative communication forms. See Table 2 for more descriptive information about the samples. 

### 4.2. Predictors of Expressive Language

The bivariate analyses (Table 3) found the same variables to be significantly related to expressive language for both groups: home literacy environment, auditory memory, oral motor skills, phonological awareness, and receptive vocabulary, in addition to the covariate nonverbal mental abilities. All relations were positive; i.e., better functioning was related to better expressive vocabulary. However, the effect sizes for auditory memory, oral motor skills, and receptive vocabulary were larger among the children with Down syndrome than among the typically developing children. In addition to the common predictors across the groups, also socioemotional functioning was significant among children with Down syndrome, but not among typical developing children.

Similar results were found when gender, parental education and nonverbal mental abilities were controlled (Table 4). Thus, better home literacy environment, auditory memory, and receptive vocabulary, in addition to the covariate nonverbal mental abilities, were significantly related to better expressive vocabulary in both groups. However, oral motor skills and phonological awareness significantly predicted only expressive vocabulary among the children with Down syndrome. As in the bivariate analyses, the effect sizes for auditory memory, oral motor skills, and receptive vocabulary were larger among the children with Down syndrome than among the typically developing children.

### 4.3. Change over Time

Both groups had significant improvements in expressive vocabulary over time. However, whereas the typically developing children improved almost 1.4 standard deviations over the two years, the children with Down syndrome improved only 0.8 standard deviations in the same period, thus improving significantly less.

## 5. Discussion

In the present study, we compared expressive vocabulary development between a group of young typically developing children and an age cohort of children with Down syndrome in the first three years of primary school who were initially matched for nonverbal mental age. We investigated what variables predicted expressive vocabulary in both groups and whether there were any significant differences between the groups in the relation between the predictors and expressive vocabulary.

Four main findings emerged from these results. (1) Both the typically developing children and children with Down syndrome showed development over time in their expressive vocabulary. The typically developing children had reliably larger changes from T1-T3 than the children with Down syndrome. (2) When gender, parental education and nonverbal mental age were controlled, both home literacy environment, receptive vocabulary and auditory memory predicted expressive vocabulary development in young typically developing children. (3) The same three variables were significant predictors of expressive vocabulary development in the children with Down syndrome. In addition, phonological awareness and oral motor skills predicted expressive vocabulary in the children with Down syndrome. (4) A comparison of predictors between the two groups showed stronger predictive values of receptive vocabulary, auditory memory, and oral motor skills in the children with Down syndrome than in the typically developing children. We discuss these main findings below.

### 5.1. Development of Expressive Vocabulary

Our results showed expressive vocabulary development over time for both typically developing children and children with Down syndrome. However, the typically developing children showed a steeper change from T1 to T3 than the children with Down syndrome. Notably, although the two groups were initially matched for nonverbal mental age, there were significant differences between the groups in expressive vocabulary at T1. The longitudinal results showed a so-called Matthews effect, with the typically developing children being the strongest group and becoming stronger. These findings were expected based on results from previous research applying clinical test measures [84,85,86]. According to Tomblin et al. [87], there is little evidence to suggest that expressive language problems occur in isolation, implying that there seems to be a multifactorial explanation of the weakness in expressive vocabulary in children with Down syndrome. This may be because expressive vocabulary involves accessing semantic knowledge in addition to phonological representations and is strongly related to word identification [88].

One explanation of the increased gap between the groups therefore may be that compared to typically developing children at the same nonverbal mental age level, children with Down syndrome usually show an initial limitation in a range of different language areas, not only expressive vocabulary. Thus, children with Down syndrome have been suggested to show compound language difficulties [17,89]. These difficulties include limitations in semantics [28,90], grammar [86,91], phonology [92], syntax [90,93] and pragmatics [94,95]. Furthermore, children with Down syndrome also tend to have weak speech sound production [89], intelligibility [90,95,96,97,98], and oral motor skills [64,90,99,100,101]. 

In addition to the overall limitations in language skills, reduced cognitive and social skills [92] are common in children with Down syndrome. Therefore, there is no single variable that explains the communication profile of children with Down syndrome [102]: their important cross-domain relationships may have long-term implications for expressive vocabulary and language processing [103,104,105]. The general weakness across areas may function as a vulnerability—according to the stress hypothesis, weak functioning in one area results in stress caused by negative experiences that interact with the vulnerability in both language aspects and other different areas, cumulating in language problems [106].

### 5.2. Predictors of Expressive Vocabulary in Typically Developing Children

Home literacy environment was the only environmental variable predicting expressive vocabulary development in the typically developing children. However, the child-related variables receptive vocabulary and auditory memory also showed reliable results. The impact of these variables were in the same direction in all the models run.

The predictive value of receptive vocabulary may be explained by the fact that infants usually understand the meanings of some words at the age of 6–8 months [107], while they usually say their first word at approximately one year of age [30]. This asynchronous developmental pattern has led to the hypothesis that the development of receptive vocabulary is the basis for the development of expressive vocabulary and that the understanding of words precedes their production [60]. However, it should be noted that research from the last decade involving fine-grained temporal analysis of individual word learning in typically developing children has shown that the progression from the comprehension of a word to the use of the same word does not necessarily reflect a linear stage-wise progression [108]. This may imply either a transactional development or that both expressive and receptive vocabulary rely on a third common variable.

One explanation for the prediction of expressive vocabulary from auditory memory may be the phonological storage hypothesis. According to this hypothesis, the close association between these two variables reflects the crucial role played by the phonological loop component of working memory in supporting the long-term phonological learning of new sound patterns [109,110]; stable phonological specifications of words are built through the abstraction of core features from temporary representations held in the phonological loop [111], which provides a direct link between temporary storage and long-term memory. However, critiques of this hypothesis have been raised, as auditory memory is often measured based on the repetition of nonwords, which are more difficult to recall than real words [112], and as there is also evidence that long-term knowledge influences processing in phonological short-term memory [113]. In this study, auditory memory was measured by several measures, including nonwords, word span and sentence memory, which may address these critical points raised against the phonological storage hypothesis. Nevertheless, it is worth noting that the association between vocabulary and auditory memory is suggested to vary throughout the developmental period [114], which may be the case for several of the other predictor variables as well.

As expected, there was also reliable effect of the home literacy environment. This finding is in line with a range of studies that have shown the impact of literacy materials and experiences on vocabulary (e.g., [48,115]). However, the home literacy variable in the present study was separated from SES and only included the frequency of the parents reading aloud to the child while some earlier studies included a mix of environmental and individual aspects in the home literacy environment variable (e.g., [116]), such as parents’ level of education, reading practice, number of picture books and the extent to which the child enjoyed book reading.

### 5.3. Predictors of Expressive Vocabulary in Children with Down Syndrome

As seen in typical developing children, the home literacy environment was the only environmental aspect measured in the study that showed any significant results in predicting expressive vocabulary development for the children with Down syndrome. In addition to auditory memory and receptive vocabulary, also phonological awareness and oral motor skills reliably predicted expressive vocabulary in the children with Down syndrome.

The relationship between vocabulary and phonology awareness has frequently been discussed in the literature, and there are several possible explanations of the relationship. One explanation is the lexical restructuring hypothesis [62,63], which views vocabulary size as a contributor to the development of the awareness of the phonological structure of new words and claims that children start to acquire new words as holistic units with global phonological characteristics [117]. These mental representations of words gradually become increasingly detailed as the child’s vocabulary size grows. Based on this hypothesis, a larger vocabulary size would gradually increase the child’s phonological skills, which in turn would result in better skills in segmenting words into small units and discerning phonologically similar words from one another. However, the influences of vocabulary and phonology may not need to be seen as mutually exclusive [118]. They may influence one another and be bidirectional in nature, with vocabulary influencing phonological acquisition and phonology influencing vocabulary. Finally, it may be that common underlying phonological skills are crucial for both auditory memory capacity and vocabulary development (c.f. [119,120]).

The finding that oral motor skills were a predictor of vocabulary development in the children with Down syndrome was expected based on previous research suggesting that weaknesses and instabilities in speech motor skills, even at a subclinical level, may disrupt the neural network that connects acoustic input, articulatory motor plans, and semantics [121]. The development of speech motor skills is commonly challenging in younger children with Down syndrome. In addition to impaired muscle control [84] and structural differences [98,122], many children with Down syndrome display characteristics of impaired oral motor planning skills [123]. Low muscle tone and structural differences influence oral motor skills, and these phenotypic characteristics affect the speaker’s intelligibility, impeding articulation and fluency [90]. Children with reduced oral motor skills struggle to pronounce and articulate new words, which can be assumed to delay expressive vocabulary development. There might also be indirect consequences; speech motor difficulties can affect the way a child is talked to and included in conversation [25,91]. According to Hart and Risley [124], children who are perceived as linguistically impaired are talked to less and in a more simplistic way. Lack of adequate linguistic input might in turn lead to reduced vocabulary development, given that input is particularly important for children with Down syndrome [25,125].

Surprisingly, the number of words the child could read did not predict expressive vocabulary development in the children with Down syndrome. In previous research, reading skills have been hypothesized to influence vocabulary in children with Down syndrome [38]. However, the reading variable in this study may have only reflected sight word reading using a logographic strategy, which may limit children’s ability to read new and unknown words and thereby restrict the number of new words children learn from reading.

The impact of the home literacy environment was consistent with results found in previous research with typically developing children that showed this variable to be important for vocabulary development (e.g., [48,115]). The frequency of parents reading aloud to the child was related to the child’s expressive vocabulary development. This finding indicates that children have the potential to learn from this stimulation and adult-child interaction, which, among other reasons, may be due to book reading being a context of joint attention that provides visual referents to words and provides a basis for word explanations and adult mediation.

### 5.4. Differences and Similarities in Predictors and Predictive Values

For the environmental variables, there were no significant differences in predictors between the typically developing children and children with Down syndrome, but the effect sizes tended to show stronger predictive values in the group of children with Down syndrome. For the child-related variables, there were significant differences between the groups for receptive vocabulary, auditory memory and oral motor skills. Receptive vocabulary and auditory memory significantly predicted expressive vocabulary in both groups, but both variables had stronger effects in the children with Down syndrome. Oral motor skills were a significant predictor of only expressive vocabulary in the children with Down syndrome.

Several aspects may explain the finding that auditory memory, vocabulary, and oral motor skills showed a stronger predictive effect on expressive vocabulary in the children with Down syndrome than in the typically developing children. One aspect is that the predictive role of predictors may vary across different stages of development. Since the children with Down syndrome did show initial weaker skills on most language measures, including expressive vocabulary, and therefore differed in their initial developmental levels, it was not unexpected that some of the predictors of expressive vocabulary differed between the two groups. Different multifactorial predictor models have been tested in typically developing children and have shown that the impact of predictors varies during the developmental period, and the predictive ability of the models tested in earlier research increased as children grew older [126,127].

Another aspect may be that the initial broad language deficit seen in children with Down syndrome may make vocabulary learning harder and result in the need for stronger support from each of these related variables in the process of the development of expressive vocabulary. A third aspect may relate to a vulnerability logic, meaning that when experiencing a problem with one aspect of language, individuals are more vulnerable to having problems with other aspects of language [106]. A fourth aspect is that the children with Down syndrome in this study had a broader range of expressive vocabulary scores. Although the typically developing children showed larger changes in expressive vocabulary, the results within the group did not vary to the same extent as that for the children with Down syndrome.

The similarities between the children with Down syndrome and typically developing children in the prediction from home literacy environment and phonological awareness show that these are core variables that predicts the development of expressive vocabulary across groups and early developmental level. However, also for these variables the predictive values were stronger in the children with Down syndrome. 

Deckers et al. [67] concluded from their parental survey that the predictors of expressive vocabulary in young children with Down syndrome (mean 57 months) seemed to resemble those that have been suggested in previous research for typically developing children. However, since Deckers et al. [67] did not include a control group of typically developing children they did not investigated whether there were any significant differences in predictors across the groups. For the common predictors that were included in both Deckers et al.’s study [67] and in ours, the prediction from vocabulary and phonological awareness were replicated within the group of children with Down syndrome. Notably, Deckers et al. [67] did not include auditory memory as a predictor variable in their study, as was the only significant predictor in our full model. Opposite to their conclusion, we will argue that not all predictors of expressive vocabulary are similar to those in typically developing children. Some positive predictor variables seem to be similar between children with Down syndrome and typically developing children, while some or at least one is not similar. In addition, the importance of the variables may not necessarily be the same for the development of expressive vocabulary in the two groups. 

The rationale for investigating differences in predictors across groups may provide a potential hypothesis to explain differences in development between groups, which in turn will provide directions for how to stimulate expressive vocabulary development. Therefore, our results add important knowledge related to predictors of expressive vocabulary development in children with Down syndrome to the literature, which is important for preparing to develop good, effective interventions that include relevant tasks. Broad-based interventions are needed to support expressive vocabulary development in children with Down syndrome.

### 5.5. Limitations

The environmental-related variables that were investigated in the present study had limited variability: the frequency with which parents read to their child, as measured by one question with five reply alternatives, and language problems in the family, measured as a dichotomous variable. A broader investigation of the home literacy environment may have found a higher effect size.

Although the present study has a substantially larger sample than is common in studies of children with Down syndrome, the sample size still imposed restriction on the number of covariates that were statistically advisable to include in the regression model. Some researchers have found that only two subjects are needed per variable [128], while others have found a much higher number of subjects is needed (e.g., approximately twenty per variable; [129]). In the present study, we focused our discussion on the results from models with 20 subjects per variable (Table 4) to minimize the risk of type II errors. However, the model including all predictors, and thus including four subjects per variable (Appendix A, Table A1), had less risk of type I errors and was therefore included for transparency. Due to being an exploratory study no adjustments for multiple comparisons are made (c.f. [130]). However, we would like to point out that even if we had adjusted the p-values with Hochberg and Benjamini [131] methods for adjustment for multiple comparisons, the group differences given in Table 4 for auditory memory and oral motor skills are still significant (*p* = 0.004 and *p* = 0.05 respectively).

## 6. Conclusions

In conclusion, both the typically developing children and children with Down syndrome showed longitudinal changes in expressive vocabulary, with more positive changes among the typically developing children than among the children with Down syndrome. Both home literacy environment, receptive vocabulary and auditory memory were significant predictors of expressive vocabulary development in the typically developing children. In the children with Down syndrome, the same three predictors in addition to phonological awareness and oral motor skills were shown to reliably predict future expressive vocabulary. There were significant differences in the predictive values between the children with Down syndrome and typically developing children on measures of receptive vocabulary, auditory memory and oral motor skills, showing a larger impact of the predictors on the development of children with Down syndrome. Additionally, for the non-significant variables, there was a tendency for the effect size to show stronger predictive value in the group of children with Down syndrome. These findings imply that children with Down are more vulnerable to risk factors known to influence expressive vocabulary than typically developing children and therefore need early, broad-based expressive vocabulary interventions.

## Figures and Tables

**Table 2 brainsci-11-00312-t002:** Descriptive information of the sample.

	Down Syndrome		Typically Developing		Group Difference		Group Difference Controlled for Gender, Parental Education Level and Nonverbal Mental Age ¹	
	(*n* = 43)		(*n* = 57)					
	Mean	*SD*	Mean	*SD*	*b*	*p*-Value	*b*	*p*-Value
Age T1 (months)	75.8	3.5	36.7	4.2	**−1.97**	**<0.001**	**−1.98**	**<0.001**
Mean parental education T1 ^1^	2.5	1.1	2.6	1.1	0.07	0.73	0.08	0.71
Nonverbal mental age T1 ^4^	12.2	5.4	12.6	4.6	0.08	0.71	0.11	0.61
Socioemotional functioning T1 ^2,4^	10.7	5.1	6.7	3.6	**−0.84**	**<0.001**	**−0.84**	**<0.001**
Auditory memory T1 ^4^	−0.5	0.5	0.3	0.8	**0.99**	**<0.001**	**0.95**	**<0.001**
Oral motor skills T1 ^3,4^	0.2	0.1	0.1	0.2	**−0.62**	**0.003**	**−0.57**	**0.005**
Phonological awareness T1 ^4^	−0.2	1.0	0.2	0.4	**0.53**	**0.008**	**0.48**	**0.01**
Receptive vocabulary T1	23.2	11.4	26.2	11.0	0.26	0.20	0.18	0.30
Expressive vocabulary T1	8.6	5.8	12.4	4.2	**0.72**	**<0.001**	**0.68**	**<0.001**
Expressive vocabulary T2	12.0	6.6	17.4	4.3	**0.90**	**<0.001**	**0.85**	**<0.001**
Expressive vocabulary T3 ¹	14.0	6.3	21.8	4.3	**1.20**	**<0.001**	**1.15**	**<0.001**

Note: Descriptive statistics for all variables are raw scores, except for auditory memory, oral motor skills and phonological awareness, for which z-values are presented. With the exception of socioemotional functioning and oral motor skills, higher scores indicate more positive outcomes. Mean standardized group differences were calculated using general linear regression analyses, in which the levels of functioning were standardized (*z*-values based on both groups) before they were entered into the models; thus, the regression coefficients (*b)* resemble Cohen’s *d* values. The multiple significance of group differences was tested using linear regression controlled for gender, mean parental education level and child’s nonverbal mental ability level (nonverbal mental abilities and education were not controlled for in their separate multiple analyses). The values of the Down syndrome group were set to zero in the analyses; thus, a positive regression coefficient indicated that the children with Down syndrome had the lowest mean score. Significant (*p* ≤ 0.05) group differences are marked in **bold text**. Descriptive information regarding categorical variables is presented in the text. ¹ *n* = 56 typically developing children. ² *n* = 41 children with Down syndrome, 55 typically developing children. ³ *n* = 37 children with Down syndrome, 55 typically developing children. ⁴ Nonverbal mental age levels were measured using the Block Design task from the Wechsler Intelligence Scale for Children-Revised (WISC-R). Socioemotional functioning was measured with the SDQ. Auditory memory was measured as the mean score of verbal short-term memory, sentence memory, and nonword repetition. Oral motor skills were measured by 14 items from the NOT-S. Phonological awareness was measured by the rhyme, initial syllable, final syllable and phoneme tests. Receptive vocabulary was measured with the BPVS. Expressive language was assessed with the Picture Naming test from the WPPSI-III.

**Table 3 brainsci-11-00312-t003:** Bivariate analysis of the predictors of expressive vocabulary over time for children with Down syndrome and typically developing children with a similar nonverbal mental age.

	Down Syndrome			Typically Developing Children with a Similar Nonverbal Mental Age			Sign. Test of Differences in Beta between Groups		
	(*n* = 43)			(*n* = 57)					
	*b*	95% CI	*p*-Value	*b*	95% CI	*p*-Value	*b*	95% CI	*p*-Value
Gender ¹	0.10	−0.33 to 0.54	0.64	0.06	−0.32 to 0.43	0.77	−0.05	−0.62 to 0.53	0.87
Nonverbal mental age T1 ²	**0.41**	**0.23 to 0.59**	**<0.001**	**0.21**	**0.03 to 0.39**	**0.02**	−0.20	−0.45 to 0.06	0.13
Mean parental education T1	0.15	−0.07 to 0.37	0.18	0.03	−0.15 to 0.22	0.73	−0.12	−0.41 to 0.17	0.42
Home literacy environment T1 ²	**0.28**	**0.04 to 0.52**	**0.02**	**0.17**	**0.01 to 0.34**	**0.04**	−0.10	−0.39 to 0.19	0.48
Language problems in family T1 ¹	−0.43	−0.94 to 0.07	0.09	−0.01	−0.43 to 0.41	0.97	0.43	−0.23 to 1.09	0.20
Socioemotional functioning T1 ²	**0.20**	**−0.00 to 0.40**	**0.05**	0.03	−0.21 to 0.28	0.80	-0.17	−0.49 to 0.15	0.29
Auditory memory T1 ²	**0.96**	**0.67 to 1.25**	**<0.001**	**0.28**	**0.14 to 0.43**	**<0.001**	**−0.68**	**−1.00 to −0.36**	**<0.001**
Oral motor skills T1 ²	**0.61**	**0.37 to 0.84**	**<0.001**	**0.16**	**0.00 to 0.32**	**0.04**	**−0.44**	**−0.73 to −0.16**	**0.002**
Phonological awareness T1 ²	**0.36**	**0.21 to 0.51**	**<0.001**	**0.31**	**0.03 to 0.60**	**0.03**	-0.05	−0.37 to 0.27	0.77
Receptive vocabulary T1 ²	**0.61**	**0.45 to 0.77**	**<0.001**	**0.34**	**0.19 to 0.48**	**<0.001**	**−0.27**	**−0.49 to −0.05**	**0.02**
Early reading skills T1 ²	0.17	−0.00 to 0.34	0.06	0.28	−0.05 to 0.62	0.10	0.12	−0.26 to 0.49	0.54
Time (change in expressive vocabulary) ¹									
T1–T2	**0.51**	**0.34 to 0.67**	**<0.001**	**0.75**	**0.60 to 0.89**	**<0.001**	**0.24**	**0.02 to 0.46**	**0.03**
T2–T3	**0.30**	**0.13 to 0.46**	**<0.001**	**0.64**	**0.50 to 0.78**	**<0.001**	**0.34**	**0.12 to 0.56**	**0.002**
T1–T3	**0.80**	**0.64 to 0.97**	**<0.001**	**1.38**	**1.24 to 1.53**	**<0.001**	**0.58**	**0.36 to 0.80**	**<0.001**

Note: All variables were originally based on raw scores. Mixed linear models were used to analyze which predictors were related to expressive language over time, as measured with the Picture Naming test. All results are fixed effects. The models included two-way interaction terms between the group and predictor variables, thus analyzing whether the difference in the regression coefficient between the groups was significant. The models were rerun with the groups reversed to determine the regression coefficient of the typically developing children and with time reversed to analyze the time at which significant changes happened. Both the dependent variable expressive language and the independent variables parental education, nonverbal mental age level, socioemotional functioning, auditory memory, oral motor skills, phonological awareness, and receptive vocabulary at T1 were standardized (*z*-values) before they were entered into the models; thus, the regression coefficients (*b*) resemble Cohen’s *d* values. The two original problem-oriented measures (socioemotional functioning and oral motor skills) were reversed before being entered into the models. Thus, all positive effect sizes within each group indicate that better functioning is related to better expressive vocabulary. A positive effect size between groups indicates a more positive relation for the typically developing children than for the children with Down syndrome. Significant (*p* ≤ 0.05) predictors within groups or significant differences in predictors between groups are marked in **bold text**. ¹ The following parameters were set to 0 in the models: male gender, no language problems in family and time 1. ² Nonverbal mental age levels were measured using the Block Design task from the WISC-R. Home literacy environment was assessed by a question asking how often parents read to their child. Socioemotional functioning was measured with the SDQ. Auditory memory was measured as the mean score of verbal short-term memory, sentence memory, and nonword repetition. Oral motor skills were measured by 14 items from the NOT-S. Phonological awareness was measured by the rhyme, initial syllable, final syllable and phoneme tests. Receptive vocabulary was measured with the BPVS. Early reading skills was measured as the number of words the child could recognize. Expressive language was assessed with the Picture Naming test from the WPPSI-III.

**Table 4 brainsci-11-00312-t004:** Multivariable analysis of the predictors of expressive language over time for children with Down syndrome and typically developing children with a similar nonverbal mental age, with gender, parental education and nonverbal mental age controlled.

	Down Syndrome			Typically Developing Children with a Similar Nonverbal Mental Age			Sign. Test of Differences in Beta between Groups		
	(*n* = 43)			(*n* = 57)					
	*b*	95% CI	*p*-Value	b	95% CI	*p*-Value	*b*	95% CI	*p*-Value
Gender ¹	−0.03	−0.42 to 0.37	0.90	0.06	−0.28 to 0.40	0.71	0.09	−0.43 to 0.61	0.73
Nonverbal mental age T1 ²	**0.41**	**0.22 to 0.59**	**<0.001**	**0.22**	**0.04 to 0.40**	**0.02**	−0.18	−0.44 to 0.07	0.16
Mean parental education T1 ²	0.14	−0.06 to 0.34	0.17	0.06	−0.11 to 0.23	0.47	−0.08	−0.34 to 0.18	0.55
Home literacy environment T1 ²	**0.25**	**0.03 to 0.47**	**0.03**	**0.15**	**−0.00 to 0.31**	**0.05**	−0.09	−0.36 to 0.17	0.48
Language problems in family T1 ¹	0.35	−0.12 to 0.81	0.15	−0.03	−0.41 to 0.35	0.87	−0.31	−0.92 to 0.29	0.30
Socioemotional functioning T1 ²	0.08	−0.11 to 0.28	0.39	0.10	−0.13 to 0.33	0.38	0.02	−0.29 to 0.33	0.90
Auditory memory T1 ²	**0.83**	**0.54 to 1.12**	**<0.001**	**0.23**	**0.09 to 0.37**	**0.002**	**−0.60**	**−0.92 to −0.28**	**<0.001**
Oral motor skills T1 ²	**0.48**	**0.25 to 0.71**	**0.001**	0.12	−0.03 to 0.27	0.11	**−0.36**	**−0.63 to −0.09**	**0.01**
Phonological awareness T1 ²	**0.30**	**0.15 to 0.44**	**<0.001**	0.18	−0.12 to 0.48	0.24	−0.12	−0.44 to 0.20	0.47
Receptive vocabulary T1 ²	**0.55**	**0.37 to 0.73**	**<0.001**	**0.30**	**0.14 to 0.45**	**<0.001**	**−0.25**	**−0.47 to −0.04**	**0.02**
Early reading skills T1 ²	0.13	−0.04 to 0.29	0.13	0.17	−0.14 to 0.48	0.29	0.04	−0.30 to 0.39	0.81
Time ¹									
T1–T2	**0.51**	**0.34 to 0.67**	**<0.001**	**0.75**	**0.60 to 0.89**	**<0.001**	**0.24**	**0.02 to 0.46**	**0.03**
T2–T3	**0.30**	**0.13 to 0.46**	**<0.001**	**0.64**	**0.50 to 0.78**	**<0.001**	**0.34**	**0.12 to 0.56**	**0.002**
T1–T3	**0.80**	**0.64 to 0.97**	**<0.001**	**1.38**	**1.24 to 1.53**	**<0.001**	**0.58**	**0.36 to 0.80**	**<0.001**

Note: Separate mixed models for each predictor were used to analyze which predictors were related to expressive language over time, as measured with the Picture Naming test. All models were controlled for gender, mean parental education level and child’s nonverbal mental age level. The models included the two-way interaction term between the predictor and group variables, thus analyzing whether the difference in the regression coefficient between the groups was significant. The model was rerun with the groups reversed to determine the regression coefficient of the typically developing children. Both the dependent variable expressive language and the independent variables parental education, nonverbal mental ability levels, socioemotional functioning, auditory memory, oral motor skills, phonological awareness, and receptive vocabulary at T1 were standardized (*z*-values) before they were entered into the models; thus, the regression coefficients (*b*) resemble Cohen’s *d* values. The two original problem-oriented measures (socioemotional functioning and oral motor skills) were reversed before being entered into the models. Thus, all positive effect sizes within each group indicate that better functioning is related to better expressive vocabulary. A positive effect size between groups indicates a more positive relation for the typically developing than for the children with Down syndrome. Significant (*p* ≤ 0.05) predictors within groups or significant differences in predictors between groups are marked in bold text. ¹ The following parameters were set to 0 in the models: male gender, no language problems in family and time 1. ² Nonverbal mental age levels were measured using the Block Design task from the WISC-R. Home literacy environment was assessed by a question asking how often parents read to their child. Socioemotional functioning was measured with the SDQ. Auditory memory was measured as the mean score of verbal short-term memory, sentence memory, and nonword repetition. Oral motor skills were measured by 14 items from the NOT-S. Phonological awareness was measured by the rhyme, initial syllable, final syllable and phoneme tests. Receptive vocabulary was measured with the BPVS. Early reading skills was measured as the number of words the child could recognize. Expressive language was assessed with the Picture Naming test from the WPPSI-III.However, when all predictors were included in the same model (Appendix A, Table A1), only auditory memory and receptive vocabulary were significant predictors among the children with Down syndrome, and none of the predictors were significant among the typically developing children. Auditory memory was the only significantly different predictor between the two groups when all variables were entered into the model.

## Data Availability

The data is available in the services for sensitive data at the University and can be available by contacting the corresponding author.

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
