# Peer review of "Differences and Similarities in Predictors of Expressive Vocabulary Development between Children with Down Syndrome and Young Typically Developing Children"

_brainsci, 2021, doi:10.3390/brainsci11030312_

Round 1

Reviewer 1 Report

Thank you for the opportunity to review this work. It is clear that the results have real practical value for practitioners and families of children with Down Syndrome.

The paper is robust in all aspects, but there are a few suggestions for minor changes that should add further clarity.

Author Response

Reply to the review report (reviewer 1)

We have prepared a revision of our manuscript brainsci-1102151 Differences in Predictors of Expressive Vocabulary Development Between Children with Down Syndrome and Young Typically Developing Children. We would like to thank the editor and the reviewers for their valuable comments on our manuscript. We have considered these comments carefully and have provided a response to each in full below (our responses are set in bold). Revisions are clearly highlighted in the manuscript by using the "*Track Changes*" function in Microsoft Word, so that changes should be easily visible to the editor and reviewers. It should be noted that this has resulted in a discrepancy in line numbers between the reviewers comments and our responses.

We want to inform you that revisions also are made in the manuscript according to comments from other reviewers, and that during the revision process we found a mistake in the manuscript regarding missing data. The analyses using mixed models had excluded participants with any missing information (listwise deletion). This listwise deletion excluded 14% of the data, even though only 1.7% of the data was missing. Even though the missing data was not completely at random (Little’s test was significant, as presented in the manuscript’s analysis section), the low rate of missing relative to the data loss with listwise deletion makes it important to use all data. We have thus now used Expectation Maximation (EM) imputation so all available data is included in the mixed models, as recommended by Heck, Thomas and Tabate (2019). This resulted in marginal changes of the results, including three predictors that were previously almost significant now being significant: socioemotional functioning for children with Down syndrome in the bivariate analyses (Table 3); home literacy environment for typical developing children in Table 4; and group differences in auditory memory in the full model (Appendix Table 1).

It is our opinion that the reviewer’s comments have contributed to strengthen the paper, and we are hopeful that it will be considered appropriate for publication in Brain Science.

On behalf of the co-authors

Kari-Anne B. Næss

  • Proofreading/minor communication errors need correction in places: e.g.

Line 28 - 'is' should be 'are'

The word is has been replaced with are in line 28.

  • Line 28-29 could be rephrased for greater clarity

The sentences in line 27-30 have been revised to: These results indicate that children with Down syndrome are more vulnerable when it comes to risk factors that are known to influence expressive vocabulary than typically developing children.

  • Line 88-89 'income; education' - is this a misplaced semi-colon?

Semicolon has been replaced with comma in line 110.

  • Line 136 ‘typically developing’ is repeated

Typically developing have been deleted in line 176.

  • Lines 132-138 – the fuller description of the Block Design test appears in line 138, after its first mention in Line 132: please swap

The fuller description of the Block Design is now moved to line 169-171.

 Section 3.2 and 3.16

Lines 169-176: ‘three separate sessions, usually on consecutive days, with one year between test points’ I am unsure how to interpret this: were the tests carried out over 3 consecutive days each year for three years (T1, 2, 3)? After reading the whole manuscript several times, I am still unclear as to when data were collected. Please give a fuller explanation (or diagram?). I am also uncertain as to which tests were collected at T2 and T3 – lines 264-8 (below) indicate that only the Picture naming Test was repeated, but Lines 172-173 says ‘clinical measures’ were repeated. Lines 264-268 ‘measured with the Picture naming tests at three different times’ – Again, is this a year apart at T1, 2, and 3?

The text in lines 216-218 has been revised and now reads: The clinical testing was administered individually in kindergartens or schools with ca 12 months between test points (T1, T2 and T3).

The research questions in lines 148-158 have also been revised to clarify what variables that are included in the different time points. These lines now read:

1)         What factors at 3 years of age predict expressive vocabulary in young typically developing children at ages 3, 4 and 5 years?

2)         What factors at 6 years of age predict expressive vocabulary in children with Down syndrome at ages 6, 7 and 8 years (first three years of primary school)?

3)         Are there any significant differences between the estimated effects of the predictors of expressive vocabulary between children with Down syndrome and typically developing children?

  • Table 3 Please reformat the columns for 95%CI+b so the data can be read horizontally

The Tables have been reformatted by the American Journal Experts.

Reviewer 2 Report

In the current manuscript, Naess et al aimed to identify predictors of expressive vocabulary development in a cohort of children with Down syndrome compared to those of typically developing children. The manuscript is as such very well-written, and all sections are well described.

I don't have any major queries pertaining to the current manuscript.

Author Response

Reply to the review report (reviewer 2)

We would like to thank the reviewer for the positive feedback on our manuscript brainsci-1102151 Differences in Predictors of Expressive Vocabulary Development Between Children with Down Syndrome and Young Typically Developing Children

We want to inform you that revisions are made in the manuscript according to comments from other reviewers (highlighted in the manuscript by using the Track Changes), and during the revision process we found a mistake in the  manuscript regarding missing data. The analyses using mixed models had excluded participants with any missing information (listwise deletion). This listwise deletion excluded 14% of the data, even though only 1.7% of the data was missing. Even though the missing data was not completely at random (Little’s test was significant, as presented in the manuscript’s analysis section), the low rate of missing relative to the data loss with listwise deletion makes it important to use all data. We have thus now used Expectation Maximation (EM) imputation so all available data is included in the mixed models, as recommended by Heck, Thomas and Tabate (2019). This resulted in marginal changes of the results, including three predictors that were previously almost significant now being significant: socioemotional functioning for children with Down syndrome in the bivariate analyses (Table 3); home literacy environment for typical developing children in Table 4; and group differences in auditory memory in the full model (Appendix Table 1).

On behalf of the co-authors

Kari-Anne B. Næss

Reviewer 3 Report

Review for Brain Science

“Differences in Predictors of Expressive Vocabulary  Development Between Children with Down Syndrome and Young Typically Developing Children "

 By Kari-Anne Næss, Johanne Ostad, and Egil Nygaard

Manuscript number : brainsci-1102151

Februabry 05, 2020

            Due to the time allowed to review the manuscript, I was unable to have my text checked by my usual proofreader. Therefore, as I am not an English-native, my written expression is very approximate. I apologize for this.

I appreciate the opportunity to review and comment on the manuscript, which describes an interesting study whose content is appropriate for the special issue of Brain Sciences on Down Syndrome. I think that this paper could be published pending some important clarification concerning the interpretation of statistical results.

- The introduction is well written, well documented and clearly states the purpose of the study. It is a strong point of the manuscript.

- The data are extensive due to the use of many predictor variables.

- The use of a longitudinal design is a very positive feature of the study.

- The discussion is also well documented and written. However, by emphasizing too much the differences between children with and without DS, the authors go far beyond the results of their statistical analyses.

- I am not an expert in mixed models. Therefore, I will not comment the choice of this approach and its implementation. On the other hand, I believe that the choices made in the statistical analyses are mainly aimed at highlighting significant between-groups differences more than similarity. It seems to me that the authors want to decrease the risk β. However they increase in an unreasonable way the risk α. For this reason, my commentaries will focus mainly on the interpretation of the statistical analyses.

The sample size is indeed important for a research involving participants with DS. However, in absolute terms (i.e. from a strictly statistical perspective), it remains quite low given the number of IVs and the number of statistical tests conducted. The authors are therefore faced with a difficult choice. They have to choose between the risk α and the risk β.

- The number of tests should lead the authors to correct the threshold α to avoid an inflation of type I errors rate. With such corrections, some differences would become insignificant at the traditional α threshold.

- In order to limit the type II errors rate, the authors favor, in their interpretation of results, the partial model including only some predictors (cf. table 4) to the detriment of the full model (cf. appendix 1). I think that the discussion should rather focus on the full model. From a scientific viewpoint, the α errors seems to me to be much more problematic than the β errors.

- There is also the question of the choice of the predictors included in the partial model and the full model. How was made this choice?

Another problem with the statistical analysis is the little attention paid to between-groups differences of b coefficients. The authors focus mainly on the statistically significant predictors in each of the groups considered in isolation and conclude that the weights of the expressive vocabulary predictors are not the same in the two groups (see Table 4). However, what should be taken into account to conclude in this sense are the results of comparisons of the b coefficients obtained by the two groups for each predictor. For example, if 6 predictors are indeed significant for participants with DS, there are only 3 significant between-group differences of b coefficients (see Table 4, partial model). It is rather on these 3 differences that the authors should emphasize in their discussion, although I think that it is the table produced in Appendix 1 which should be discussed

To summarized, the statistical analysis strategy adopted by the authors aims to emphasize differences: by not correcting the alpha thresholds for multiple comparisons, by focusing the discussion on the partial model to the expense of the full model, by minimizing, in their discussion, the low number of between-group differences of b coefficients. I believe that the data do not support the conclusion that the predictors of expressive vocabulary are different for the two groups. In my opinion, this is due to a widespread habit in psychology, that of wanting to absolutely put in evidence significant effects. Of course, this also applies to the intellectual disability field in general, and to the syndromic approach in particular. The risk is to magnify differences at the expense of developmental similarities.

            To conclude, I think the authors could present their results in the form of two figures showing the expressive vocabulary level at T1, T2 and T3. One figure for the partial model (but in explaining the choice made among variable) and one for the full model. This will give the reader a more concrete idea of the developmental trajectories of the two groups. Table 4 could be put in an appendix to avoid redundancy with the corresponding figure. The authors should also emphasize, in their discussion and on the basis of their full model, that there are no between-group differences in b coefficients and, therefore, that the groups do not differ concerning the predictors of expressive vocabulary acquisition. Finally, they could indicate, as a limitation of the study, the fact that the absence of differences could be due to a lack of statistical power. Of course, a significant part of the discussion should be changed, particularly with regard to the theoretical and practical implications of nonsignificant between group differences.

One last remark. The authors point out that the development of expressive vocabulary is faster in typical children than it is among children with DS. Is this a specificity of language development of children with DS or is it due to their slower cognitive development? Indeed, the cognitive level in T2 and T3 is no longer the same for the two groups given the intellectual deficiency of children with DS.

Round 2

Reviewer 3 Report

Review for Brain Science

Februabry 19, 2020

“Differences in Predictors of Expressive Vocabulary  Development Between Children with Down Syndrome and Young Typically Developing Children "

By Kari-Anne Næss, Johanne Ostad, and Egil Nygaard

Manuscript number : brainsci-1102151-Revised

Thank you for the modifications provided by the authors. Below, a few additional remarks about their letter and the content of the revised article. As I am not an English-native, my written expression is very approximate. I apologize for this.

The question of the compromise to be found between alpha and beta errors is at the heart of the manuscript. Indeed, if one wants to avoid type I error, one increases the risk of type 2 error. I am refractory to alpha errors. Indeed, alpha errors are almost never corrected because of the very low number of replications of research studies done in psychology. This is a proven fact (Makel et al. 2012). However, without replications, no possibility of self-correction, especially in the case of publication of false-positive results (Ioannidis, 2012).

The present study, even if the authors consider it to be "exploratory", will most certainly be cited without underlining its "exploratory" character. This "exploratory" result will soon become a "definitive" result, as researchers in the developmental disability field value significant effects, especially when it comes to "profiling" the psychological characteristics of children with intellectual disability. Publication bias as well as citation bias against papers reporting no differences do exist, it is a reality.

I adhere to the idea of many researchers as to the very pejorative nature of Type 1 error on the development of scientific knowledge. Hence the call by many researchers for the alpha threshold to be significantly lowered in scientific research (Benjamin et al., 2017; Johnson et al., 2017).

I therefore suggest that the authors place as much emphasis on their two models (partial and complete) and point out much more clearly the fact that the results can be interpreted in terms of differences as well as similarities. Simply putting the full model in an appendix tends to devaluate its significance, to present it as "secondary". Giving the reader the opportunity to make a choice should consist of giving them all the information without implicitly or explicitly leading them to favor one option over the other.

            I also think that the authors should specify the basis on which they have designed the partial model and the full model. Why put certain variables in one model and not the other? What led to this allocation?

One last remark: you should write "Hochberg" instead of "Hockberg".

References

Benjamin, D. J., Berger, J. O., Johannesson, M., Nosek, B. A., Wagenmakers, E. J., Berk, R., Bollen, K. A., Brembs, B., Brown, L., Camerer, C., Cesarini, D., Chambers, C. D., Clyde, M., Cook, T. D., De Boeck, P., Dienes, Z., Dreber, A., Easwaran, K., Efferson, C., ... Johnson, V. E. (2018). Redefine statistical significance. Nature Human Behaviour, 2(1), 6-10.

Ioannidis, J. P. A. (2012). Why Science Is Not Necessarily Self-Correcting. Perspectives on Psychological Science, 7(6), 645–654.

Johnson, V. E. (2013). Revised standards for statistical evidence. PNAS Proceedings of the National Academy of Sciences of the United States of America, 110(48), 19313–19317.

Makel, M., Plucker, J., & Hegarty, B. (2012). Replications in psychology research: How often do they really occur? Perspectives in Psychological Science, 7, 537–542.

Review for Brain Science

Februabry 19, 2020

“Differences in Predictors of Expressive Vocabulary  Development Between Children with Down Syndrome and Young Typically Developing Children "

By Kari-Anne Næss, Johanne Ostad, and Egil Nygaard

Manuscript number : brainsci-1102151-Revised

Thank you for the modifications provided by the authors. Below, a few additional remarks about their letter and the content of the revised article. As I am not an English-native, my written expression is very approximate. I apologize for this.

The question of the compromise to be found between alpha and beta errors is at the heart of the manuscript. Indeed, if one wants to avoid type I error, one increases the risk of type 2 error. I am refractory to alpha errors. Indeed, alpha errors are almost never corrected because of the very low number of replications of research studies done in psychology. This is a proven fact (Makel et al. 2012). However, without replications, no possibility of self-correction, especially in the case of publication of false-positive results (Ioannidis, 2012).

The present study, even if the authors consider it to be "exploratory", will most certainly be cited without underlining its "exploratory" character. This "exploratory" result will soon become a "definitive" result, as researchers in the developmental disability field value significant effects, especially when it comes to "profiling" the psychological characteristics of children with intellectual disability. Publication bias as well as citation bias against papers reporting no differences do exist, it is a reality.

I adhere to the idea of many researchers as to the very pejorative nature of Type 1 error on the development of scientific knowledge. Hence the call by many researchers for the alpha threshold to be significantly lowered in scientific research (Benjamin et al., 2017; Johnson et al., 2017).

I therefore suggest that the authors place as much emphasis on their two models (partial and complete) and point out much more clearly the fact that the results can be interpreted in terms of differences as well as similarities. Simply putting the full model in an appendix tends to devaluate its significance, to present it as "secondary". Giving the reader the opportunity to make a choice should consist of giving them all the information without implicitly or explicitly leading them to favor one option over the other.

            I also think that the authors should specify the basis on which they have designed the partial model and the full model. Why put certain variables in one model and not the other? What led to this allocation?

One last remark: you should write "Hochberg" instead of "Hockberg".

References

Benjamin, D. J., Berger, J. O., Johannesson, M., Nosek, B. A., Wagenmakers, E. J., Berk, R., Bollen, K. A., Brembs, B., Brown, L., Camerer, C., Cesarini, D., Chambers, C. D., Clyde, M., Cook, T. D., De Boeck, P., Dienes, Z., Dreber, A., Easwaran, K., Efferson, C., ... Johnson, V. E. (2018). Redefine statistical significance. Nature Human Behaviour, 2(1), 6-10.

Ioannidis, J. P. A. (2012). Why Science Is Not Necessarily Self-Correcting. Perspectives on Psychological Science, 7(6), 645–654.

Johnson, V. E. (2013). Revised standards for statistical evidence. PNAS Proceedings of the National Academy of Sciences of the United States of America, 110(48), 19313–19317.

Makel, M., Plucker, J., & Hegarty, B. (2012). Replications in psychology research: How often do they really occur? Perspectives in Psychological Science, 7, 537–542.
